# Primitive Reflex Activity in Relation to the Sensory Profile in Healthy Preschool Children

**DOI:** 10.3390/ijerph17218210

**Published:** 2020-11-06

**Authors:** Anna Pecuch, Ewa Gieysztor, Marlena Telenga, Ewelina Wolańska, Mateusz Kowal, Małgorzata Paprocka-Borowicz

**Affiliations:** 1Physiotherapy Department, Faculty of Health Sciences, Wrocław Medical University, 50-355 Wrocław, Poland; anna.pecuch@student.umed.wroc.pl (A.P.); marlena.telenga@student.umed.wroc.pl (M.T.); mateusz.kowal@umed.wroc.pl (M.K.); malgorzata.paprocka-borowicz@umed.wroc.pl (M.P.-B.); 2Department of Pediatrics, Division of Pediatrics and Rare Disorders, Wrocław Medical University, ul. Bartla 5, 51-618 Wrocław, Poland; ewelina.wolanska@student.umed.wroc.pl

**Keywords:** sensory profile, primitive reflexes, preschool children, neurodevelopment

## Abstract

The presence of active primitive reflexes (APRs) in healthy preschool children can be an expression of immaturity in the functioning of the nervous system. Their trace presence may not significantly affect the quality of child functioning. They may also undergo spontaneous and complete integration within the stages of child development. However, a higher level of active reflexes and their significant number can disturb sensory-motor development and lead to additional problems in a child’s motor activities, social life, and education. The main purpose of this study was to examine the types of sensory disorders noticed by parents of children, if any, that accompany the presence of active primitive reflexes. The study was conducted in a group of 44 preschool children (aged 4–6 years). The sensory profile of children was determined using Child Sensory Profile Cards, and Sally Goddard-Blythe tests were used to measure their primitive reflexes. The coefficient of determination (R-squared) indicated that the level of reflex activity was most strongly associated with sensory disorders such as dyspraxia, sensory-vestibular disorders, and postural disorders, at a level of *p* < 0.005. The obtained research results show that the examination of non-integrated reflexes might be a screening tool for children of preschool age. Knowledge of the subject of reflexes and their impact on sensory-motor functions may contribute to more accurate diagnoses of the causes of problems and higher effectiveness of possible therapy.

## 1. Introduction

Healthy preschool children may present varying levels of sensory and motor development [1]. Sensory-motor problems of a mild degree may have a chance to be integrated on their own as the child grows. The problem, however, is serious for a preschool or elementary school-age child with numerous significant sensory-motor disorders that do not disappear spontaneously, or begin to increase with age [2]. Parents and teachers often notice these disorders in children in the form of difficulties in maintaining balance and clumsiness during physical activities, or problems in learning and social functioning [3]. Determining the origin of these problems is often crucial for the selection of appropriate remedies and, consequently, improving the quality of the child’s functioning. One factor that may negatively influences the functioning of a child over the period of infacy is the presence of active primary reflexes (also known as ‘persistent primary reflexes’ (PPR)) [4]. However, the occurrence of non-integrated primary reflexes is common among preschool children [1,5,6]. Authors dealing with this topic report that they occur in about 90% of healthy preschool children and about 55% of early school-age children. In most preschool children (65%), these reflexes occur at a low level of activity (visible as trace motor reactions). When a child has only one or two slightly non-integrated reflexes, it can be assumed that, with the maturation in the functioning of the nervous system and further development of the child, these reflexes will undergo spontaneous integration. A larger number of non-integrated reflexes and their high level of activity might be a reason for problems that disturb proper sensory-motor development and social functioning. Active primary reflexes can make a child’s posture and motor skills appear clumsy. Children with APRs may have a specific gait picture [7] or learning difficulties [8]. Instinctive reflexive movements caused by active reflexes can affect both the large and small motor skills of children.

Primary reflexes develop during fetal life. They are responsible for stereotypical and involuntary motor reactions in response to internal and external stimuli [9]. Their control center is located in the brainstem. They facilitate child delivery and help children during their first moments after birth, and enable normal psychomotor development in the first months of life [10]. Reflex activity persists up to several months after birth and should spontaneously disappear (integrate into the nervous system) as higher motor skills begin to develop [4,10]. This takes place when a primary reflex, e.g., the Moro reflex, which is an instinctive defensive reaction and allows the child to take the first breath of life, is replaced by the Strauss (startle) reflex, which can continue for the remainder of a person’s life [9]. The development of primary reflexes in fetal life is possible due the parallel development of equivalent structures. After birth, primary reflexes are triggered by stimuli from the vestibular system and other sensory channels. An example of this is the asymmetrical tonic neck reflex (ATNR), which can be induced by the movement of the head. The tonic stimulus, which is the rotation of the neck, provokes a deflection of the upper and lower limbs on the occipital side of the body, and trunk rotation [11]. On the facial side, the limbs straighten. ATNR in a three-month-old girl is shown in Figure 1. 

The activity of primary reflexes beyond the biological period of proper occurrence may influence proper sensory-motor development [5,7,8]. If primary reflexes are not integrated completely during the spontaneous development of the nervous system, they will be constantly provoked by the same stimuli. These reactions are less strong and stereotypical compared to a newborn. However, even a slight degree of reflex activity may result in muscle tensions that appear in the child’s body and cause involuntary motor reactions. In stressful or new situations, an active reflex will cause a primitive motor response such as a slight muscle tone or visible motion. The response of the body depends on the degree and the number of APRs. 

The tensions and movements generated by the activity of reflexes may hinder functioning and can cause uncertainty about the child’s own bodily reaction in various everyday situations (during motor activities such as cycling or physical games with peers). Fear of sudden changes in the environment may cause a child to withdraw and abandon new intellectual and motor skill-oriented challenges [1,9]. Reflex activities can cause learning difficulties in preschool and elementary school [12,13,14]. They may also cause difficulties in acquiring age-related mobility skills. Early experiences of “standing out” from a group of peers in terms of mobility, manual abilities, and difficulties experienced by a child in learning didactic material can cause frustration and lead to secondary problems [15,16,17,18,19]. Disorders in sensory-motor development can affect the cognitive and emotional development of a child, and thus the child’s contact with peers and social functioning [20,21].

The main purpose of this work was to address the issue of the existence of a relationship between the presence of active primitive reflexes in healthy children and sensory disorders noticed by their parents.

## 2. Materials and Methods

This work is part of the PRACS (Primitive Reflexes and All Children Sphere) project, which is aimed at investigating primitive reflexes and their impact on the motor, sensory, and cognitive development in preschool and school-age children. One paper showing the impact of ATNR on the symmetry of a child’s gait is now available [7]. A second article, which has featured in a series of publications as part of the project, shows a child’s perception of their developmental difficulties in relation to the adult assessment of those difficulties [3].

In a group of preschool children, both quantitative and qualitative levels of the occurrence of selected primitive reflexes were examined. The parents of the children were also asked to complete questionnaires regarding their child’s sensory profile. The research was approved by the Ethics Committee of the Wrocław Medical University, approval number KB-626/2018. It was carried out in accordance with the Helsinki Declaration. The children’s parents were kept informed of the purpose and non-invasive nature of the research. Each gave written consent for their child to participate in the study.

### 2.1. Participants

The data were collected from 44 children (28 girls and 16 boys). The mean age of the group was 4.8 (± 0.98) years. Subjects’ characteristics is shown in Table 1. Exclusion criteria included a medical or pedagogical diagnosis of special needs. Therefore, children diagnosed with motor or intellectual disabilities, or children with a psychological and pedagogical opinion issued by a psychological counseling center confirming the need to support their educational process, were excluded from the research.

### 2.2. Assessment of Reflex Activity

Tests developed by Sally Goddard-Blythe in 1996 at the Institute of Neurophysiological Psychology (INPP) in the UK were used to assess the prevalence of primitive reflexes in children. The above tests were included in the “Test Set for Children Aged 4–7” in a script for people who have completed a course enabling them to diagnose the presence of surviving primary reflexes in preschool and early-school children [22].

Tests were conducted to examine the integration of tonic reflexes during extension (EXT) and flexion (FLX) variants for symmetrical tonic neck reflex (STNR) and tonic labyrinthine reflex (TLR). Items for testing symmetrical tonic neck reflex and tonic labyrinthine reflexes are shown in Figure 2, Figure 3, Figure 4 and Figure 5. These reflexes are studied by changing the position of the head. If the child has fully integrated tonic reflexes, then the change in head position should not cause any movement in the lower and upper limbs, and torso. If the movement of the head causes movement and changes the position of the body then an incomplete integration of this reflex in the child is confirmed.
The symmetrical tonic neck reflex (STNR) was tested in two variants. When the child’s head was extended, it was observed whether there were symptoms such as extending the upper limbs at the elbows, sitting on the heels, or trunk movements (Figure 2). After flexing the child’s head, it was observed whether the elbows were bent, the pelvis lifted or knees extended, and the general behavior of the child. The therapist also assessed the intensity level of the current compensations (Figure 3).The tonic labyrinthine reflex (TLR) test is presented in Figure 4. Compensations such as hand or entire upper limb movements, climbing on toes, disturbance, or loss of balance were observed during the test for TLR extension. When performing the test for TLR flexion, attention was paid to the appearance of compensation in the form of fist-clenching, knee deflection, disturbances, or loss of balance. The test position is presented in Figure 5.The asymmetrical tonic neck reflex (ATNR) test for the right (R) and left (L) side is presented in Figure 6 and Figure 7. During the ATRN study, it was observed whether the rotation of the head to the right or left was accompanied by a change in the position of the upper limbs (elbows and shoulders bending), and movement in the trunk and pelvis.The Galant reflex (truncal incurvation reflex) test and the Palmar grasp test for the right (R) and left (L) side are presented in Figure 8 and Figure 9. The Palmar grasp and Galant reflex were caused by stimulation of the skin. The Palmar grasp was induced by a spatula movement on the palm (submaximal pressure). A diagnosis of Galant reflex was carried out by “drawing” a vertical line along the paravertebral region from the thoracic to the lumbar region using a thumb. In the case of current active skin reflexes, we can observe the presence of movement after the action of the stimulus in the area of stimulation (after the stimulus was activated, children can rub the stimulated area).

The last reflex examined in one variant was the Moro reflex, as presented in Figure 10. The Moro reflex was tested by leaning back on the therapist’s arm after closing the eyes and bending the head back first. It was observed whether the child bends the knees, throws the arms to the side, controls the movement during the tilting back, or performs other co-movements. Children who do not have an active Moro reflex are eager to perform this test task.

The STNR, ATNR, and Galant reflex tests were carried out in four point-kneeling positions. The Moro reflex, TLR, and Palmar reflex were tested in a standing position. While examining the STNR, ATNR, TLR, and Moro reflex, it is recommended to have the child close the eyes before starting head movements. The skin reflexes (Palmar grasp and Galant reflex) were tested with the child’s eyes open.

When assessing the ATNR, STNR, TLR, and Moro reflex, attention was paid to the quantitative and qualitative compensations within the upper and lower limbs, in addition to the shoulders, the pelvic girdles, and the trunk. The presence of an active primitive reflex may also be evidenced by a change in the rhythm of breathing, frowning, pursing lips, and emotional reactions. All of the compensations present during the performance of the tests were taken into account when assessing the degree of activity of a given reflex. Reflexes were assessed on a five-step scale from 0 to 4. A 0 meant a complete lack of reflex (full integration), and 1—low activity, 2—medium activity, 3—high activity, 4—maximum activity. The total sum of points in the examination reflexes was 40. The sum of the points obtained during the examination of all reflexes was converted to the level of reflex activity on a scale from 0 to 4. The scale is shown in Table 2.

### 2.3. Sensory Profile Assessment

The Child Sensory Profile Cards issued by the Laboratory of Psychological and Pedagogical Tests (a questionnaire for preschool and early-school children) [23] were used to assess the children’s sensory profile. Parents answered questions about the presence or lack of appropriately described behavior in their child. To assess the level of sensory disorders, the sum of points obtained in each subgroup was calculated. The results are presented as percentages. The authors listed eight subgroups of sensory integration disorders in line with the Ayres concept. These are:Dyspraxia. Defined as an impairment of motor coordination, problems with balance, distance assessment, and a sense of one’s own body in space.Tactile sensitivity. Manifested by avoiding the touch of other people and textures along with an exaggerated pain response (even to gentle touch).Visual-auditory-vestibular seeking. Characterized by an excessive interest in glowing/spinning/sound-emitting objects, the desire by the child to generate sounds/noise, and the enjoyment of spinning around one’s body axis.Sensory-vestibular seeking. A penchant for scuffling and heavy objects, the need for pressure, and intentional colliding with objects.Postural problems. Manifested by reduced coordination-equivalent abilities, frequent stumbling, and rapid fatigue, difficulty crossing the body centerline.Vestibular hyperactivity. Characterized by the avoidance of rapid movement, fear of positional changes, and falls.Olfactory-sensory-taste seeking. Characterized by an excessive interest in strong flavors and olfactory stimuli, as well as the desire to touch, and put objects in the mouth to examine their texture and taste.Auditory hyperactivity. Manifested by the avoidance of and distraction by even the soft or otherwise inconspicuous sounds (e.g., the sound of an air conditioner or a watch) and responding with fear to the sounds of animals, radio, etc.

The authors considered the final result at the level of 25% as the cut-off point indicating an increased number of disorders.

Diagnosing disorders of the sensory integration processes requires a detailed interview with parents and carrying out appropriate tests and trials. The diagnosis of sensory disorders in the child can be made based on the Californian, SIPT (Senssory Integration and Praxis Test), or EASI (Evaluation in Ayres Sensory Integration) tests [24,25]. The applied tool, i.e., children’s sensory profile, allows for a preliminary assessment of children in terms of potential disorders and drawing the parents’ attention to their child’s problems in a given area [20]. It is an excellent screening tool, but only allows for an initial assessment of the child. In addition, the results of the sensory profile draw our attention to abnormalities in children without identified disturbances in the processes of sensory integration.

### 2.4. Statistics

Statistic analysis was performed using IBM SPSS Statistics version 25 (IBM Corp., Armonk, NY, USA). Descriptive statistics were computed for all variables. Arithmetic means and standard deviation were calculated. The distribution was determined by the Shapiro–Wilk test. It was found that the distribution of the data significantly deviates from a normal distribution. For the presented results, linear regression calculations were performed (estimated by the least squares method) and the R-squared determination factor was calculated. The test was conducted to examine the correlation between the reflex activity level results and sensory profile results for individual categories. Differences between girls and boys and comparison between age groups were tested by a Mann–Whitney U test. Comparisons in age groups were performed using an ANOVA test. The level of significance for purposes of interpretation of the analyses, *p* < 0.05 was adopted.

## 3. Results

### 3.1. Reflex Activity Level in Examined Group

The sum of the points obtained during the examination of all reflexes was converted to the level of reflex activity. The results show that 2.3% of the examined children had no reflex activity. A significant proportion (84.1%) of examined children had a low (40.9%) or medium (43.2%) level of reflex activity. A high level of reflex activity was observed in 13.6% of all subjects. The maximum level of reflex activity was not demonstrated by any child included in the study. The results of the reflex activity level in the examined group of preschoolers are shown in Figure 11.

### 3.2. Results of APR Examination

The most common reflexes in the group of examined preschoolers were TLR EXT (observed in 90.9% of children), ATNR L (84.1% of children) and ATNR P (84.1% of children), and Moro reflex (observed in 84.1% of children). Slightly fewer children presented STNR EXT reflex (75%) and STNR FLX (42%). Skin reflexes, i.e., Galant reflex and Palmar grasp, occurred less frequently. The reflexes that were most often expressed to the maximum degree were the Moro reflex (70.5%) and TLR EXT (27.3%). The maximum degree for the STNR FLX and the Palmar grasp in both the left and right hand was not present in any of the children. The results expressed as a percentage are shown in Figure 12 and Figure 13.

### 3.3. Sensory Profile Results

Of the 44 children tested, 11.4% had a result equal to or above 25% of the mean value of the sum of all disorders, whereas in 6.8% of children, no sensory disorders were found. Furthermore, 81.8% of children had a result in the range of 2–24% of the mean value of the sum of all disorders. The average values obtained in each category are presented in Table 3.

### 3.4. The Level of Reflex Activity and Sensory Disorders

The applied linear regression demonstrated that the level of reflex activity (the total points obtained during the examination) is positively correlated with the occurrence of the individual sensory disorders demonstrated by the sensory profile. The results are shown in Table 4.

### 3.5. Differences between Girls and Boys in the Level of Reflex Activity and Sensory Profile

To determine whether girls and boys differed in the level of reflex activity, the Mann–Whitney U test was carried out. The results are shown in Table 5. There were statistically significant differences between the final results obtained by boys and girls. Statistical differences between girls and boys were found for TLR EXT reflex and the Palmar grasp R. There was also a significant difference in the sum of points obtained in the examination of all reflexes. Girls obtained a significantly lower mean number of points than boys for the above-mentioned reflexes and in the final result. 

Statistically significant differences were also observed between the results of boys and girls in two categories of the sensory profile. In tactile hyperactivity, girls obtained a significantly higher score than boys. Boys obtained a significantly higher result in sensory-vestibular seeking than tested girls. The results are shown in Table 6.

### 3.6. Differences between the Activity of Reflexes and the Results of the Sensory Profile for Individual Age Groups

To check whether age was related to APR examination and sensory profile results, an ANOVA test was carried out. Children were divided according to age into groups of 4-,5-, and 6-year-olds. There were no statistically significant differences in the results obtained in the APR examination and the sensory profile with regard to age.

## 4. Discussion

The presence of primary reflexes in a child in the first year of life affects the child’s motor and cognitive-emotional development [10]. Accordingly, the presence of active primitive reflexes (APRs) in trace form may have an impact on the sensory-motor development and intellectual spheres of the preschool child [26,27].

The literature provides copious information concerning the occurrence of primary reflexes in people with a damaged nervous system, e.g., due to cerebral palsy or post-stroke [4]. There is still little research on the prevalence of APRs and their importance for the development and functioning of healthy populations. The majority of the published work is concerned with children with learning disorders and attention deficits [8,17,18,19], but a few papers also exist about the co-existence of APRs with postural and motor problems. Research has also been conducted that shows that APRs may have an impact on motor skills [1], gait [7], and scoliosis processing [28,29]. However, studies show that the prevalence of primary reflexes at the level of minor or moderate disorders is a common occurrence in preschool and early school-age children [2,5,6]. Comparing the results of APR examination of preschool and elementary school-age children, it is apparent that the number of reflexes and their degree of expression decreases with the child’s age. Goddard-Blythe [26] states that the diminished and rarely expressed reflexes may undergo spontaneous integration with age. However, the presence of more than one or two significantly expressed reflexes can cause neurodevelopmental delay and may have an impact on functioning throughout the child’s life [15].

Gieysztor et al. [1], who studied the presence of ATNR, STNR, and TLR among healthy children aged 4–6 years, report that these residual degree reflexes occur in the majority (i.e., approximately 65%) of the preschool children population. In 25% of children, the reflexes were at the most significant and highest level. This study showed the presence of these three tonic reflexes in 89% of subjects [2]. Due to the examination of three additional reflexes (Palmar, Galant, and Moro reflex) in our study, the presence of APRs was found in 98% of examined children. In the majority (84.1%), the level of reflex activity was low or moderate. In our studies, no child exhibited the highest level of reflex activity. This observation seems to confirm the fact that, in a population of healthy children, there is no increase or very strong presence of APRs, because it would indicate serious problems in the functioning of the nervous system. According to Gieysztor et al. [1], among the three tonic reflexes they examined in the preschool population, the most common reflex is ATNR L and the least common is STNR FLX. Our research partially confirms this fact. There is also a frequent occurrence of the TLR EXT reflex, as reported by Madejewska [6], who stated that it was the most common reflex occurring among children aged 4–7 years. Gieysztor et al. [1] show that TLR EXT and ATNR L occur at a maximum degree more often than STNR FLX. Our more recent study also confirms this relationship. The Moro reflex was also often expressed to the maximum degree. It should be understood that the Moro reflex expressed in a preschool- or school-aged child at the maximum level is only a trace of the reflex that occurs in newborns. Its expression at the maximum level in preschool children is a strong reluctance to take the test, the rejection of hands while moving, tilting backward, and bending the knees. Sometimes the examination is accompanied by a strong emotional reaction. Due to the fact that the test for checking the presence of the active Moro reflex is a strong sensory experience (the child closes their eyes during the test, tilts their head back, and falls backward into the hands of the therapist), there is a risk of over-recognition of this reflex in the youngest children.

Ahn et al. [20], who researched preschool children using a Short Sensory Profile (a questionnaire assessing parents’ perceptions of behavioral responsiveness of children to sensations), showed that the disorders of sensory processing occur in 5% to 10% of healthy children. In our research, the sensory profile was also determined by the parents. The mean result for a sensory profile above 25% (which may indicate sensory processing disorders) was found in 11% of the subjects. Our research shows that for the initial assessment of the occurrence of sensory disorders in a child, one can use tools such as the Sensory Profile, in which the parent answers questions about possible sensory disorders in their child. For a detailed assessment of sensory disorders in a child, specially prepared tests should be used, i.e., those conducted by a qualified sensory integration therapist.

The prevalence of APRs in a significant proportion of the preschool child population is the motivation at address the question: to what extent, if at all, do APRs have an impact on various spheres of functioning? Our research shows that the percentage of the population of preschool children in whom parents observe an increased number of sensory problems (in children who had a result of 25% in the Sensory Profile study) is about 11%. We observed a high level of APR in 13.6% of the examined children. In 81.8% of cases, the sensory problems observed by the parents are at the level of mild disorders (children who had results from 2 to 24% in the Sensory Profile test). Similarly, the presence of low and moderately expressed APR occurs in 84.1% of the study group. These results suggest that special attention, and perhaps appropriately selected therapy, should be given to children in whom we observe significantly and numerously expressed reflexes. Children with slightly or moderately present reflexes should be only under observation and have another examination after one year. This would make it possible to determine the percentage of the population for which slightly and moderately expressed reflexes disappear spontaneously, and whether their presence may be the cause of noticeable developmental problems in the future (e.g., in preschool). It also seems important to establish the norms of the occurrence of APR in the population of healthy children. Therefore, if we observe the phenomenon of a trace form of primary reflexes at a low or moderate level in a significant portion of preschool children (in about 65–84% of healthy preschool children), then perhaps such a phenomenon should be considered normal. The same conclusion was suggested by Gieysztor in her study [2]. The occurrence of primary reflexes to a low or moderate degree in such a large number of the population contributes to the fact that this phenomenon cannot be a significant pathology. Perhaps due to the fact that there is a phenomenon of neuromuscular memory, the traces of the primitive reflexes may extend beyond infancy. It also appears justified to conduct long-term studies on the occurrence and spontaneous integration of reflexes with age.

According to the children’s sensory profiles, which were completed by their parents, problems such as dyspraxia and postural problems, were one of the categories that most strongly correlated with a high level of reflex activity. No studies have been found that would conclusively confirm the relationship between sensory disturbances and the activity of primary reflexes. However, the literature provides information about the relationship between the activity of primary reflexes and sensory-motor disorders concerning, inter alia, balance, coordination, and cognitive processes [1,5,12,13]. The results of our study show that, although individual reflexes do not seem to be frequently correlated with individual sensory disorders, their accumulation significantly correlates with them. The R-squared determination coefficient indicated that dyspraxia, sensory-vestibular disorder, and postural disorders were most strongly associated with reflex activity levels. These results show that children with equivalent coordination and movement difficulties have more numerous and higher APRs than children without difficulties mentioned above. The direct reason for this is the fact that primary reflexes examined in children in our study are related to the work of structures responsible for balance, proprioception, and touch. This fact is also confirmed by the results presented by Gieysztor [1], who examined the relationship between the motor skills of preschool children and the occurrence of tonic reflexes (ATNR, STNR, and TLR). The movement skill of preschool children was examined using the Motor Proficiency Test (MOT 4–6). The final result of this test significantly correlated with the result obtained in the examination of tonic reflexes. The MOT test assessed, among other things, coordination abilities and equivalency in children in movement tests such as balancing, catching, and throwing or jumping. Children perform less well in movement tasks such as catching a ball, cycling, or balancing, which is due to poor posture and poor eye-hand coordination [13,24,26]. The relationship between APRs and the way of moving and positioning the pelvis while walking in preschool children was described by Gieysztor et al. [7]. The common occurrence of non-integrated primaryreflexes in the child population was also described by Grzywniak [13] and Bilbilaj et al. [8]. They pointed to the problem of APRs among children who had already reached school age. The presence of moderately and strongly persistent primary reflexes can be an important reason for apparent but often inexplicable learning difficulties. Children with persistent reflexes have problems with crossing the visual midline, which prevents them from effectively learning how to to read and write (the effect of ATNR activity), sitting calmly on a bench (the effect of Galant reflex), or copying notes from a blackboard to a notebook (the result of STNR and TLR) [9]. Children with APRs are also emotionally unstable (they can be excessively explosive, irritable, or extremely withdrawn and distrustful of their own abilities) and create educational difficulties (they often get hysterical while parents attempt to understand their behavior) [17,18,19]. Masgutowa [30] also indicates the problem of coexistence and dependence of sensory integration disorders and the presence of primitive reflexes with speech disorders. This facts confirms that balance-coordination and congregational disorders are among those whose presence may often be accompanied by primitive reflexes.

An increased presence of primary reflexes may be noticed by parents or teachers as clumsiness, problems with balance and coordination, inadequate emotional behavior, or difficulties with concentration. Based on our own and other authors’ studies [15,31] dealing with the coexistence of various developmental problems with active primitive reflexes, it has been demonstrated that a proper diagnosis of the causes of a child’s problems is necessary for the process of planning the therapy of children with sensory-motor disorders. Focusing only on symptoms may result in an incorrect selection of therapeutic measures and, as a consequence, ineffective therapy. The reflex tensions in the child’s body as a consequence of the trace form of APR presence will prevent these problems from being completely solved, despite the use of equivalent exercises or other elements of sensory integration therapy (e.g., tactile desensitization). The effect of therapy can be disproportionately low compared to the child’s and the therapist’s efforts because the causes of sensory-motor problems, which may be non-integrated primary reflexes, will still be present in the child’s “body.” To integrate reflexes at an age other than physiologically predicted during this period, properly targeted therapy may be needed. Selected exercises performed individually with a child give the child a “second chance” to integrate reflexes that are expressed to a degree that makes it difficult to function. Children with only a few minimally expressed reflexes may benefit from properly selected and conducted group exercises. Simple balance and coordination exercises, and the improvement of motor skills, are also able to provide stimuli that will allow the child to reach neuromotor maturity and integrate primary reflexes. In children with highly expressed primary reflexes, the Sally Goddard-Blythe method of reflex extinction can be used, however, it should be performed by a qualified therapist.

The data were compared between girls and boys. In the TLR EXT reflex, Palmar reflex on the right side, and total sum of points in the reflex examination, girls showed a higher degree of integration (in these test mean results in girls were significantlly lower than in boys). Examined boys achieved higher scores in one category (sensory-vestibular seeking) of the sensory profile, indicating they showed greater problems in a given area. In the research of Gieysztor [1] and Madejewska [6] on the presence of primitive reflexes in preschool children and their motor skills, the girls also obtained results indicating a higher level of reflex integration and better motor skills. However, their results were not statistically significant. Comparing these data, it can be noted that preschool-age girls are characterized by a higher level of sensorimotor development than boys, which is a commonly accepted fact.

Results obtained after analyzing the data based on age may suggest that, for preschool children, the level of reflexes and sensory disturbances is not necessarily age-related. The sum of the points obtained in the study and the level of the reflex activity in all groups was similar. According to Gieysztor’s study [2], differences in the level of reflex integration can be observed between preschool and school children. The spontaneous gradual integration of reflexes occurs with the age of children and reaching school maturity. However, high levels of reflexes are also observed in some school children. Grzywniak [31] reports that in children with learning difficulties, reflexes do not seem to integrate spontaneously, but increase with age. It is, therefore, reasonable to pay special attention, including long-term observation, in the case of children who have been screened for numerous and strongly expressed primitive reflexes and sensory-motor disorders.

## 5. Conclusions

Although parents often perceive their child’s problems, they find it difficult to establish the causes. It is often difficult for therapists to pinpoint the exact cause of various sensory disorders. Sensory problems (including dyspraxia and postural disorders) occurring in children are usually dismissed as the child’s clumsiness. The reasons for clumsiness and sensory-motor problems could be active primary reflexes. The existence of a correlation between the presence of a higher level of APRs and sensory disorders observed by parents shows that examination of the child’s primary reflexes may be a good screening tool to determine the child’s development.

The presence of a low or mild level of active primary reflexes (this phenomenon affects a significant part of the population of preschool children) does not seem to cause significant sensory-motor problems in a child. It appears important for child therapists that numerous reflexes in a significant or very intensified form can considerably impede the functioning and inhibit the proper development for some children. The results obtained from the research on the activity of primitive reflexes show that correct knowledge in this area might help determine the cause of a child’s problems and plan appropriately targeted therapy. The elimination of the cause of problems, which may be the activity of persistent reflexes, will result in a higher effectiveness of other therapeutic activities.

## Figures and Tables

**Figure 1 ijerph-17-08210-f001:**
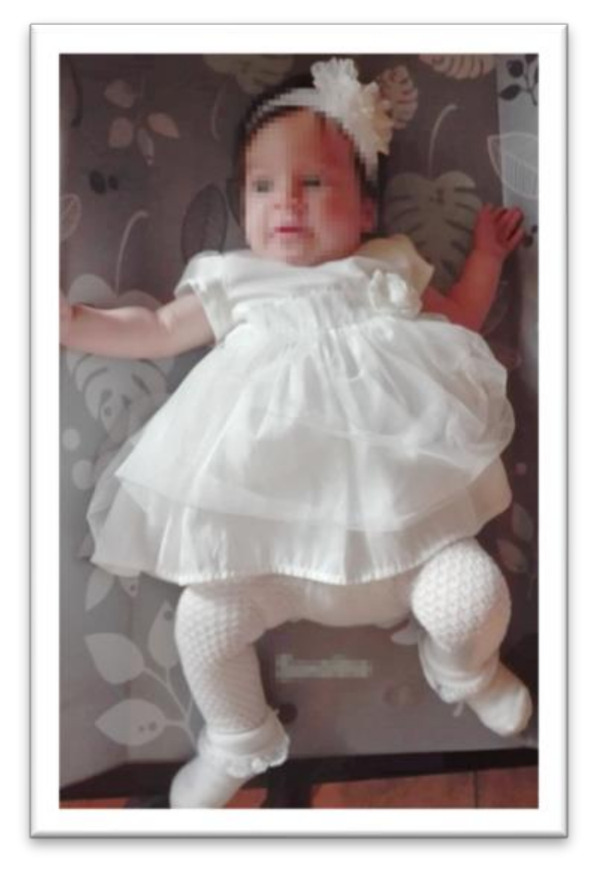
Asymmetrical tonic neck reflex (ATNR) expression in an infant.

**Figure 2 ijerph-17-08210-f002:**
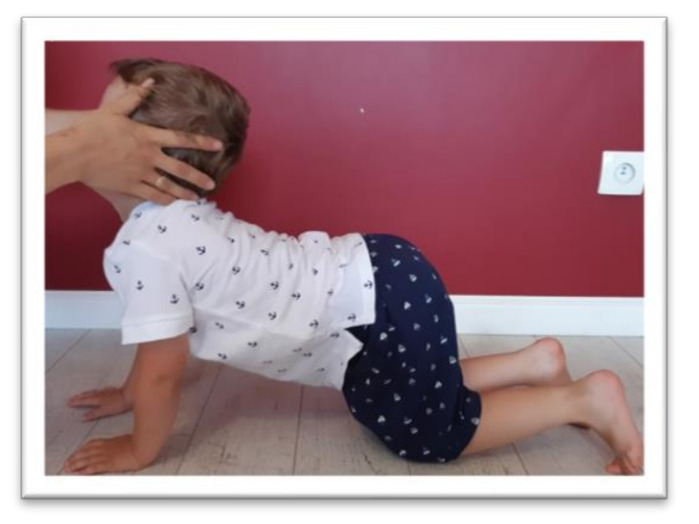
Position for the examination of symmetrical tonic neck reflex (STNR) extension.

**Figure 3 ijerph-17-08210-f003:**
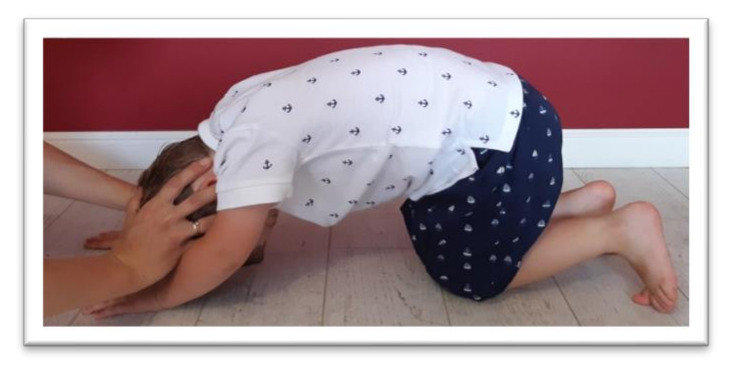
Position for the examination of STNR flexion.

**Figure 4 ijerph-17-08210-f004:**
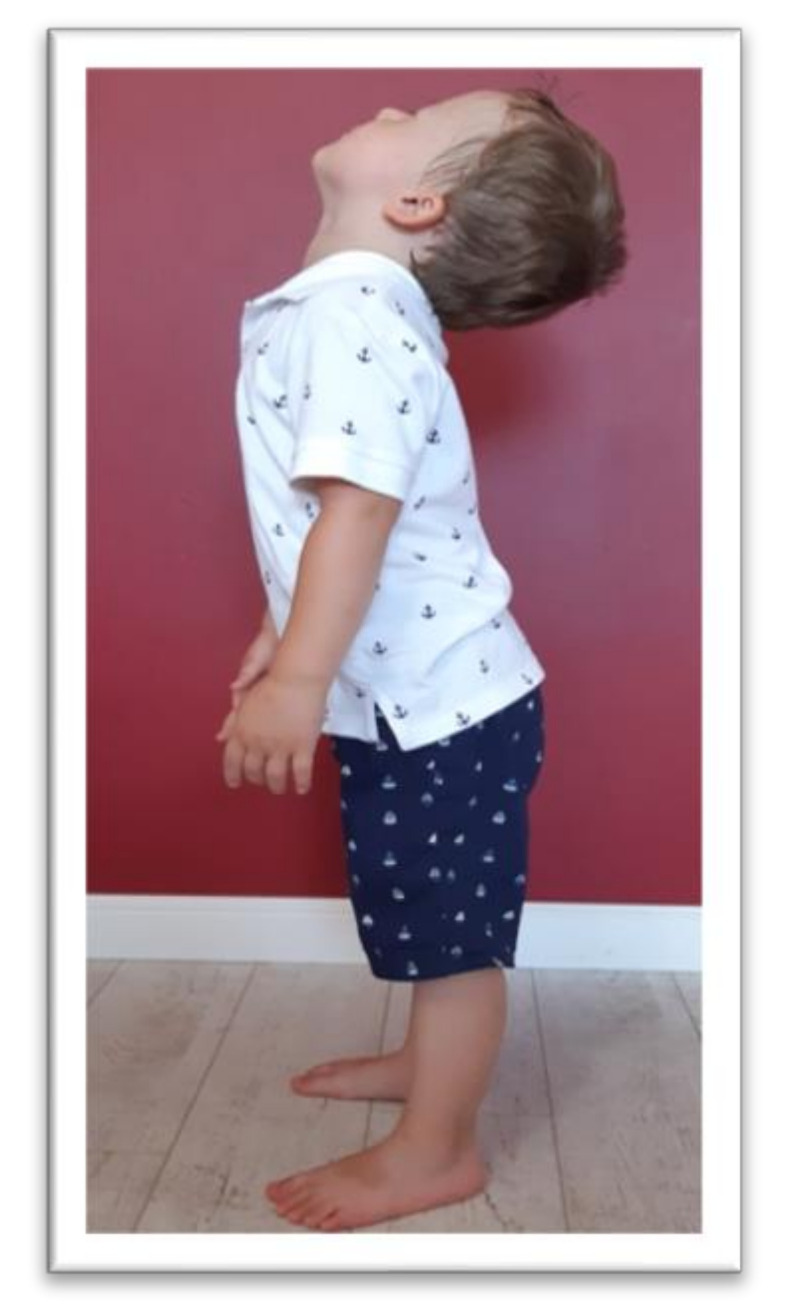
Position for the examination of tonic labyrinthine reflex (TLR) extension.

**Figure 5 ijerph-17-08210-f005:**
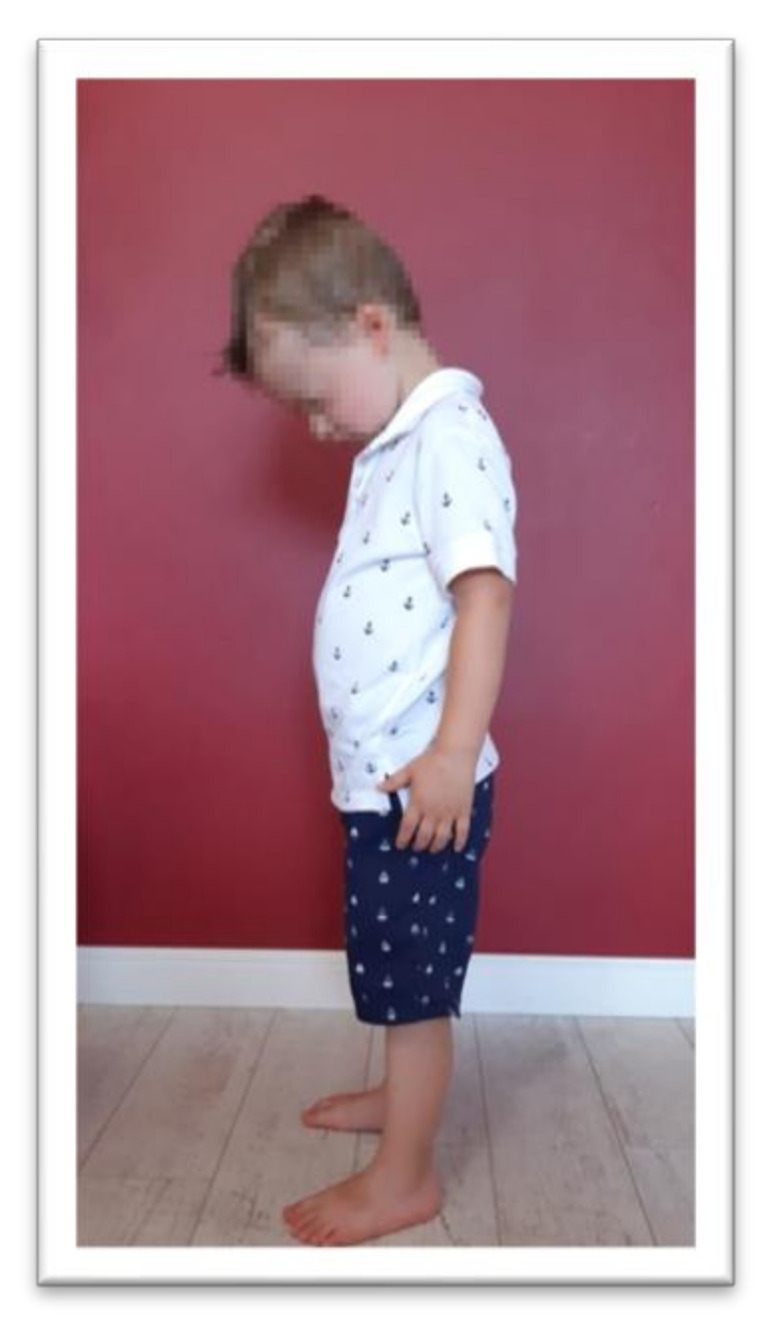
Position for the examination of TLR flexion.

**Figure 6 ijerph-17-08210-f006:**
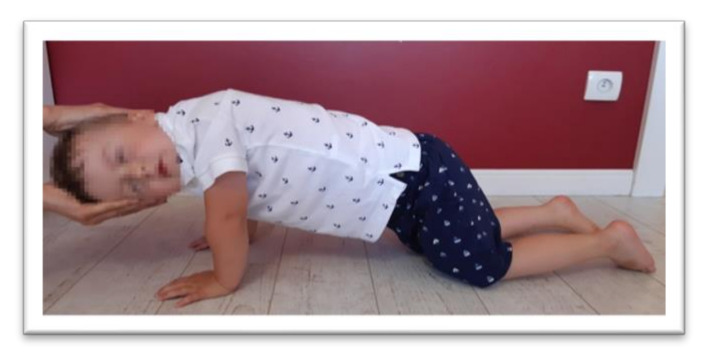
Position for the examination of ATNR for the left side.

**Figure 7 ijerph-17-08210-f007:**
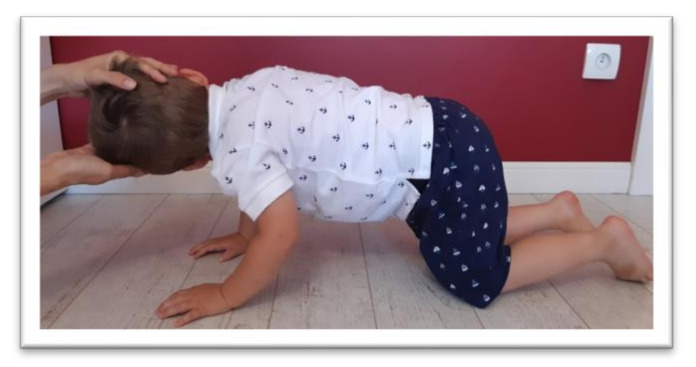
Position for the examination of ATNR for the right side.

**Figure 8 ijerph-17-08210-f008:**
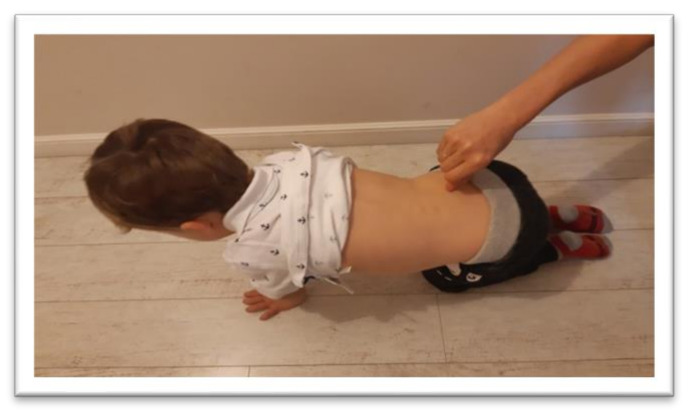
Position for the examination of Galant reflex for the right side.

**Figure 9 ijerph-17-08210-f009:**
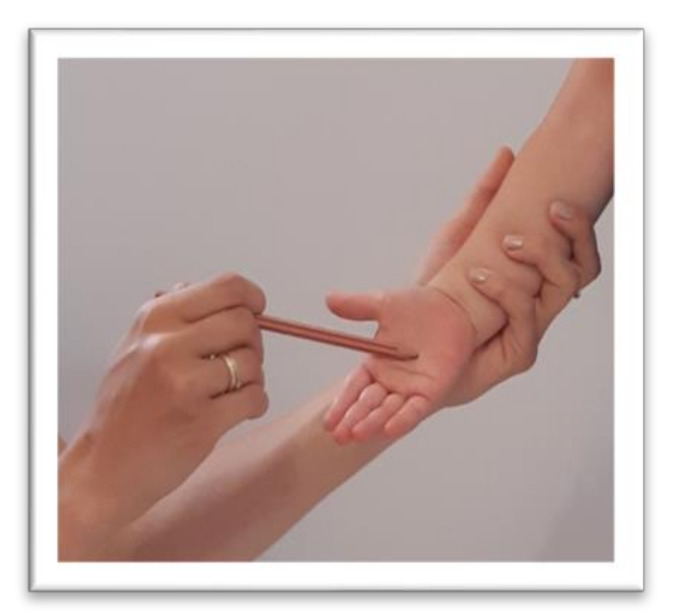
Position for the examination of Palmar reflex for the right side.

**Figure 10 ijerph-17-08210-f010:**
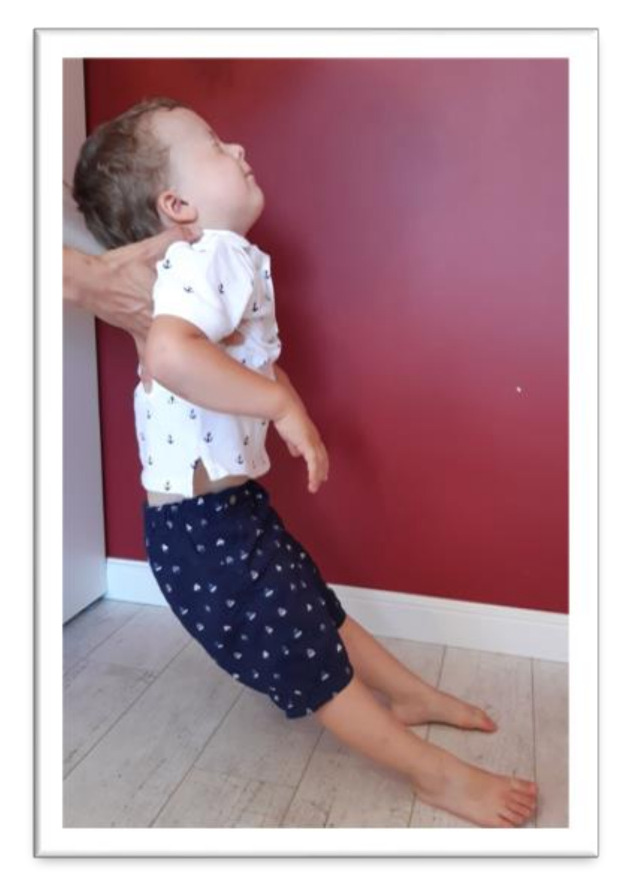
Position for the examination of the Moro reflex.

**Figure 11 ijerph-17-08210-f011:**
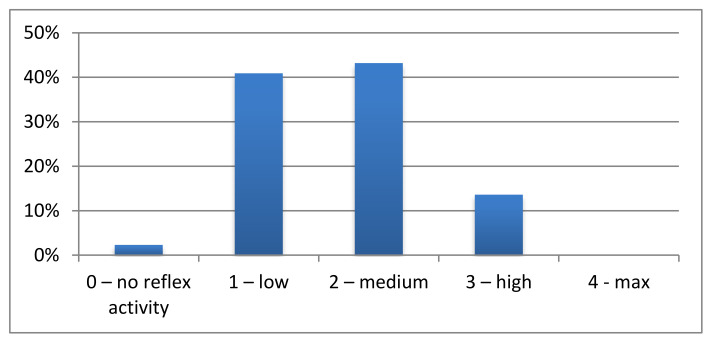
The results of the level of reflex activity.

**Figure 12 ijerph-17-08210-f012:**
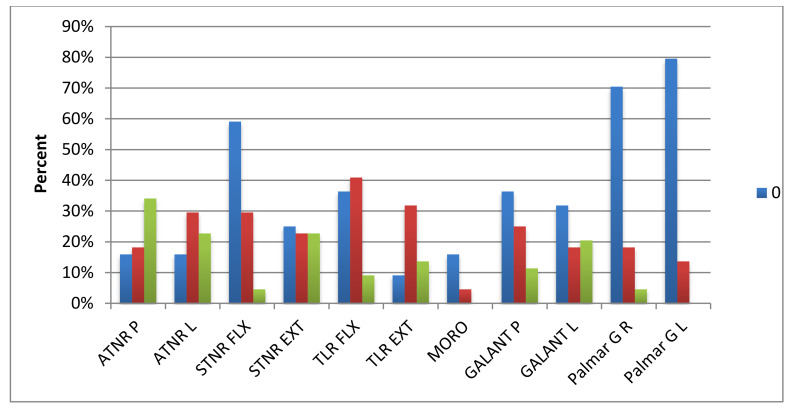
Results of the reflex test on a 0–2 scale.

**Figure 13 ijerph-17-08210-f013:**
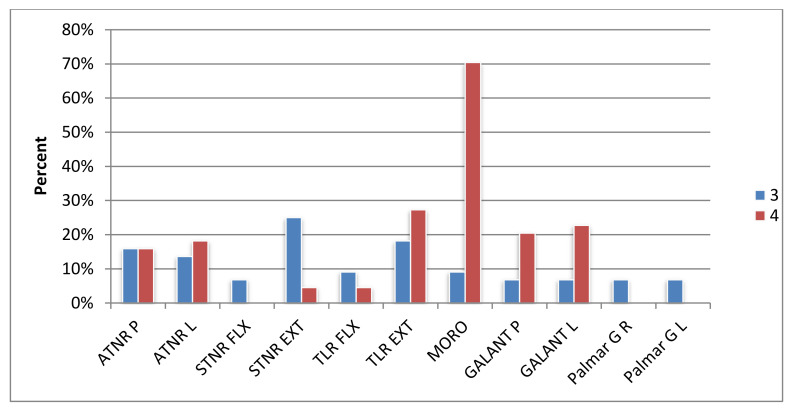
Results of the reflex test on a 3–4 scale.

**Table 1 ijerph-17-08210-t001:** Subjects’ characteristics.

Name of Group	Age	Girls	Boys
Four-year-olds	4 years–4 years and 11 months	15	6
Five-year-olds	5 years–5 years and 11 months	7	5
Six-year-olds	6 years–6 years and 11 months	6	5

**Table 2 ijerph-17-08210-t002:** The degree of primitive reflex integration scale.

Final Score in APR Examination	Level of Reflex Activity
0–3	0—no activity
4–15	1—low
16–25	2—medium
26–35	3—high
36–40	4—max

**Table 3 ijerph-17-08210-t003:** Results of the sensory profile.

Sensory Disorders	Mean Value	Max Value	SD
Dyspraxia	13%	60%	15%
Tactile hyperactivity	19%	71%	17%
Visual- auditory-vestibular seeking	15%	100%	22%
Sensory-vestibular seeking	19%	100%	25%
Postural problems	8%	50%	13%
Vestibular hyperactivity	8%	50%	13%
Olfactory-sensory-taste seeking	1%	33%	5%
Auditory hyperactivity	14%	80%	19%

The minimum value in each subgroup was 0%.

**Table 4 ijerph-17-08210-t004:** Coefficient of determination (R-squared) of sensory disorders with the corresponding activity level of reflexes.

Sensory Disorders	Level of Reflex Activity	*p*-Value
Dyspraxia	0.771	0.0005
Visual-auditory-vestibular seeking	0.312	0.0362
Sensory-vestibular seeking	0.642	0.0031
Postural problems	0.456	0.0037
Olfactory-sensory-taste seeking	0.261	0.0006

**Table 5 ijerph-17-08210-t005:** Differences between girls and boys in the results of the examination of reflexes.

Reflex	Girls, Mean ± SD	Boys, Mean ± SD	U	p-Value	η2
STNR EXT	1.6 ± 1.3	1.6 ± 1.05	223	0.5	0.0
STNR FLX	0.5 ± 0.9	0.69 ± 0.68	177.5	0.13	0.029
TLR EXT	1.75 ± 1.2	3.1 ± 1.2	106	0.002 *	0.188
TLR FLX	0.96 ± 1.05	1.2 ± 1.2	202	0.3	0.007
ATNR L	1.7 ± 1.3	2.5 ± 1.3	171	0.1	0.038
ATNR R	1.8 ± 1.15	2.3 ± 1.4	168.5	0.09	0.042
Galant L	1.8 ± 1.5	1.5 ± 1.5	197	0.26	0.01
Galant R	1.5 ± 1.6	1.6 ± 1.5	209.5	0.36	0.003
Palmar grasp L	0.2 ± 0.6	0.6 ± 0.99	164	0.07	0.049
Palmar grasp R	0.25 ± 0.7	0.9 ± 0.99	132.5	0.013 *	0.113
Moro reflex	3.1 ± 1.5	3.2 ± 1.5	212.5	0.39	0.002
Total reflex points	15.14 ± 6.9	18.99 ± 6.3	149	0.034 *	0.076
Level of reflex activity	1.5 ± 0.7	1.9 ± 0.75	180	0.119	0.026

SD-standard deviation; U-Mann-Whitney test results; p-level of significance; η2 – effect size; * *p* < 0.05.

**Table 6 ijerph-17-08210-t006:** Differences between girls and boys in the results of the sensory profile.

Sensory Disorders	Girls, Mean ± SD	Boys, Mean ± SD	U	p-Value	η2
Dyspraxia	11% ± 14%	15% ± 17%	201.5	0.29	0.007
Tactile hyperactivity	23% ± 18%	12% ± 13%	143.5	0.026 *	0.088
Visual-auditory-vestibular seeking	16% ± 20%	15% ± 25%	211.5	0.39	0.002
Sensory-vestibular seeking	12% ± 19%	30% ± 29%	144.5	0.027 *	0.086
Postural problems	7% ± 10%	9% ± 16%	216.5	0.43	0.001
Vestibular hyperactivity	1% ± 6%	0% ± 0%	216	0.43	0.001
Olfactory-sensory-taste seeking	8% ± 14%	8% ± 10%	197.5	0.26	0.01
Auditory hyperactivity	14% ± 19%	15% ± 18%	213	0.4	0.002

SD-standard deviation; U-Mann-Whitney test results; p-level of significance; η2 – effect size; * *p* < 0.05.

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
