# Peer review of "Primitive Reflex Activity in Relation to the Sensory Profile in Healthy Preschool Children"

_ijerph, 2020, doi:10.3390/ijerph17218210_

Round 1
Reviewer 1 Report
The paper is interesting but have several issues that should be resolved, as indicated below.
Abstract:
- Why the acronym PPR was used for “active primitive reflexes”? This comment also applies to the Introduction.
Materials and Methods:
- The pictures of the infants tested should be blurred (at least a part of the face).
- As for statistics, have you checked for the normality of the data (e.g., with a Shapiro-Wilk Test) and proceeded with the analysis accordingly?
Results:
- Figure 11 seems to be incomplete. Please, check.
Conclusion:
- This section should be re-numbered as 5. Conclusions.
- The first sentence appears awkward. The same applies to the one at lines 419-420. Please, check.
- English language and grammar should be revised throughout the manuscript. The quality is really low. As such, some typos are also present.
Author Response
Dear Reviewer
I am sending my manuscript again under the changed title ,,Primitive reflex activity in relation to the sensory profile in healthy preschool children”.
When applying the changes, I took into account all comments and remarks of reviewers. I included information about this in the reply to the reviewers and in the file with the corrected text. In addition, the article has been revised by a professional language translator. I hope that this fact significantly improved the quality of my text, as it was one of the main objections of all reviewers.
Yours sincerely
Anna Pecuch
Respond to comments
Abstract:
Comment:
Why the acronym PPR was used for “active primitive reflexes”? This comment also applies to the Introduction.
Response:
Acronim PPR was also used in a previous article in this project's series (in Gieysztor’s, E.Z. Pecuch, A. et.al. Pelvic Symmetry Is Influenced by Asymmetrical Tonic Neck Reflex during Young Children’s Gait). Its full meaninig was Persisten Primary Reflexes. I wanted this acronym to stay the same, but I use the term Active Primary Reflexes more than Persistent so I replaced the acronym PPR with APR.
Materials and Methods:
Comment:
The pictures of the infants tested should be blurred (at least a part of the face).-
Response:
Children's faces are now blurred
Comment:
As for statistics, have you checked for the normality of the data (e.g., with a Shapiro-Wilk Test) and proceeded with the analysis accordingly?
Response:
The data distribution was checked with a Shapiro-Wilk Test. It is different from normal, therefore, U Mann Withney and ANOVA Kruskala-Wallisa tests were used to compare children between genders (U Manna Withney) and in terms of age (ANOVA), as you suggest. Thank you for this valuable attention. Information about this was included in the text.
Results:
Comment:
Figure 11 seems to be incomplete. Please, check.
Response:
The drawing was actually incomplete. There must have been an error loading the file into the publishing system.
Conclusion:
Comment:
This section should be re-numbered as 5.
Response:
I corrected it.
Comment:
The first sentence appears awkward. The same applies to the one at lines 419-420. Please, chec
Response:
The sentence has been corrected.
The entire article has been thoroughly checked once again and has been proofreaded by a native.

Reviewer 2 Report
I have reviewed this article two months ago and although I can see that the authors have made significant changes in the resubmitted article it still requires further changes. As I said in my earlier review this paper requires extensive English language editing. It is very hard to read and the contents often appear to be delivering mixed messages!
I began a line by line analysis:
Line 2: Primitive reflex activity
Abstract
Line 14: PPR actually stands for ‘primitive preborn reflexes’ - you cannot call it one thing and then give it a different acronym
Line 14: can be
This is going to have to be rewritten because of what I have alluded to above. You are calling them active primitive reflexes but using the acronym PPR – this is very confusing to the reader. The English used here is also rather stilted and does not always convey what I believe you are trying to say. As I said in my earlier review I strongly recommend you get an English speaking person to edit this paper.
Introduction
Lines 33-53: Again this needs considerable work on the English language presentation. I tried to do this on the PDF document – but then decided that it would be better if you co-opted an English speaker to edit this. I will give you an example:
Healthy preschool children may present varying levels of sensory and motor development. Sensory-motor problems of a mild degree, may have a chance to be integrated on their own, as the child grows up. The problem, however, becomes serious when a preschool or elementary school-age child has many and significant sensory-motor disorders, and they do not disappear spontaneously or begin to increase with age.
The remaining part of the introduction also need these minor edits – which as a reviewer I cannot do.
Line 56-57: They enable the child to experience the delivery of their first movements after birth, as well as enabling normal psychomotor development in the first months of life.
I have read this through several times – but have become somewhat overwhelmed with the amount of data and how you are describing it. Even your conclusion is somewhat vague.
Finally I decided to give a broader overview:
I can see that you have a great deal of data – which could – if more succinctly described offer a very useful screening device for children. However I believe that this paper must be written much more clearly and carefully in order to avoid over-diagnosis of childhood problems which could ultimately lead to over-treatment. Paediatric therapists have been here before in the ‘70s and ‘80s with Doman-Delarcato and I personally would not like to see us going there again.
I have tried to deduce from the numbers the differences you have found in the different age groups but I am still somewhat confused. You do seem to have a great deal of information about these children but I believe it needs to be written with a great deal more clarity before it is publishable

Author Response
Dear Reviewer
I am sending my manuscript again under the changed title ,,Primitive reflex activity in relation to the sensory profile in healthy preschool children”.
When applying the changes, I took into account all comments and remarks of reviewers. I included information about this in the reply to the reviewers and in the file with the corrected text. In addition, the article has been revised by a professional language translator. I hope that this fact significantly improved the quality of my text, as it was one of the main objections of all reviewers.
Yours sincerely
Anna Pecuch
Reviewer 2
Comment:
I have reviewed this article two months ago and although I can see that the authors have made significant changes in the resubmitted article it still requires further changes. As I said in my earlier review this paper requires extensive English language editing. It is very hard to read and the contents often appear to be delivering mixed messages!
Response:
The article has been rethought and improved both in terms of content and language.
Comment:
Line 2: Primitive reflex activity
Response:
Corrected in the text
Abstract
Comment:
Line 14: PPR actually stands for ‘primitive preborn reflexes’ - you cannot call it one thing and then give it a different acronym
Response:
Acronim PPR was also used in a previous article in this project's series (in Gieysztor’s, E.Z. Pecuch, A. et.al. Pelvic Symmetry Is Influenced by Asymmetrical Tonic Neck Reflex during Young Children’s Gait). Its full meaninig was Persisten Primary Reflexes. I wanted this acronym to stay the same, but I use the term Active Primary Reflexes more than Persistent so I replaced the acronym PPR with APR.
Comment:
Line 14: can be
Response:
Corrected in the text
Comment:
This is going to have to be rewritten because of what I have alluded to above. You are calling them active primitive reflexes but using the acronym PPR – this is very confusing to the reader. The English used here is also rather stilted and does not always convey what I believe you are trying to say. As I said in my earlier review I strongly recommend you get an English speaking person to edit this paper.
Introduction
Comment:
Lines 33-53: Again this needs considerable work on the English language presentation. I tried to do this on the PDF document – but then decided that it would be better if you co-opted an English speaker to edit this. I will give you an example:
Healthy preschool children may present varying levels of sensory and motor development. Sensory-motor problems of a mild degree, may have a chance to be integrated on their own, as the child grows up. The problem, however, becomes serious when a preschool or elementary school-age child has many and significant sensory-motor disorders, and they do not disappear spontaneously or begin to increase with age.
The remaining part of the introduction also need these minor edits – which as a reviewer I cannot do.
Response:
Changes have been applied. The entire article has been thoroughly checked once again and has been proofreaded by a native
Comment:
Line 56-57: They enable the child to experience the delivery of their first movements after birth, as well as enabling normal psychomotor development in the first months of life.
Response:
Corrected in the text
Comment:
I have read this through several times – but have become somewhat overwhelmed with the amount of data and how you are describing it. Even your conclusion is somewhat vague.
Finally I decided to give a broader overview:
I can see that you have a great deal of data – which could – if more succinctly described offer a very useful screening device for children. However I believe that this paper must be written much more clearly and carefully in order to avoid over-diagnosis of childhood problems which could ultimately lead to over-treatment. Paediatric therapists have been here before in the ‘70s and ‘80s with Doman-Delarcato and I personally would not like to see us going there again.
I have tried to deduce from the numbers the differences you have found in the different age groups but I am still somewhat confused. You do seem to have a great deal of information about these children but I believe it needs to be written with a great deal more clarity before it is publishable.
Response:
Thank you for your valuable comments on the text. The revised version of manuscript contains new text fragments explaining that we are also far from talking about the occurrence of primary reflexes as a pathology, since we observe this phenomenon in 80-90% of preschool children. We are more inclined to the conclusion that perhaps the occurrence of primary reflexes to a slight or even moderate degree in children of preschool age may be the norm. Since in most preschool children we observe the presence of low or moderate reflexes, it may be necessary to consider which of them require specific targeted therapy and which can only remain under observation. In those who have a lot of and significantly expressed reflexes, parents observe more sensory problems and it is important. This can be taken into account in the therapeutic planning process. I also wrote about it in the discusion and conclusion.
Due to comments from other reviewers with a suggestion to change the applied statistical tests, the results were presented slightly differently and it seems that they are now presented much more clearly.

Reviewer 3 Report
Thank you for the opportunity to review your manuscript.
I have some comments that I think will strengthen the paper.
- Are there any references for the statements at the beginning of the intro?
e.g., lines 33-40 do not have any references
- What disorders were included under “Exclusion criteria was the physician or pedagogical diagnosis about special needs.” This may be important information with respect to the different sensory profiles compared.
- In the statistical section, you repeat the same phrase twice, “Descriptive statistics were computed for all variables. Descriptive statistics were computed for all variables.”
- Should boys and girls be compared using chi-square analysis instead?
- Why do three t-tests to compare age instead of non-parametric analysis for K groups instead?
Author Response
Dear Reviewer
I am sending my manuscript again under the changed title ,,Primitive reflex activity in relation to the sensory profile in healthy preschool children”.
When applying the changes, I took into account all comments and remarks of reviewers. I included information about this in the reply to the reviewers and in the file with the corrected text. In addition, the article has been revised by a professional language translator. I hope that this fact significantly improved the quality of my text, as it was one of the main objections of all reviewers.
Yours sincerely
Anna Pecuch
Respond to comments
Reviever 3
Comment:
Are there any references for the statements at the beginning of the intro? e.g., lines 33-40 do not have any references
Response:
New references have been addend
Comment:
What disorders were included under “Exclusion criteria was the physician or pedagogical diagnosis about special needs.” This may be important information with respect to the different sensory profiles compared.
Response:
The information is provided in the text.
Comment:
In the statistical section, you repeat the same phrase twice, “Descriptive statistics were computed for all variables. Descriptive statistics were computed for all variables.”
Response:
This has been corrected.
Comment:
Should boys and girls be compared using chi-square analysis instead?
Response:
We established the Shapiro-Wilk tetsem distribution which was included in the text.
On the advice of a statistician, we made this comparison using the Mann Whitney U test.
Comment:
Why do three t-tests to compare age instead of non-parametric analysis for K groups instead?
Response:
We did this comparison again as suggested by the ANOVA test
Round 2
Reviewer 1 Report
Changes made according to my comments.
Author Response
Dear Reviewer
The text has been checked and spelled corrected once again. All the suggestions regarding the linguistic corrections of the other reviewers were taken into account.
Kind regards
Anna Pecuch

Reviewer 2 Report
Significant improvements - I am now happy to recommend publication after minor revisions.
Please see changes on pdf.

Author Response
Dear Reviewer
The text has been checked and spelled corrected once again. All the suggestions regarding the linguistic corrections and other were taken into account.
Kind regards
Anna Pecuch

This manuscript is a resubmission of an earlier submission. The following is a list of the peer review reports and author responses from that submission.
Round 1
Reviewer 1 Report
Dear authors,
Thank you for the manuscript and all the effort put into it. However the current study is ambiguous in in its title and purpose. For e.g. the title is ''The importance of diagnosis and therapy of primary reflexes for therapeutic advances in sensory processing disorder''. The title does not make sense; you cannot diagnose primary reflexes. It is unclear what the title is trying to say. The purpose of ''to show that disorders of sensory integration in preschool children can be accompanied by non-integrated primitive reflexes'' is disparate from the title.
More critically, the methodology is flawed and needs re-thinking. The current inclusion criteria is flawed; simply recruiting 50 non-specific children and to answer questions on sensory processing disorders is not acceptable.
The interpretation of the results and thereby conclusions do not reflect that of the methodology. It is unclear how each can be led from the methodology.
The literature review, presentation and writing style also fall short of academic rigour. In particular, lines 15-18 state ''Reflexes diagnosis and integration therapy in children is as popular of a procedure as sensory integration therapy but it seems reasonable to introduce this procedure as a standard examination of children with sensorimotor disorders.'' This sentence does not make sense.
All in all, the research purpose and methodology needs re-thinking and re-working. It is unclear what the current results mean.
Reviewer 2 Report
The paper is potentially interesting as it deals with a somewhat novel, albeit extremely significant topic from a clinical perspective. The main drawback of this paper is related to the quality of presentation, that should be significantly improved. Also, some parts need to be revised, including:
- Introduction: some more literature about sensory disturbances assessed by both subjective and objective measurement methods should be cited
- Materials and Methods:
- the reference to Ethical Committee approval number should be included
- no statistical approach was explained
- Results: statistics is not presented if we exclude the short sentence at the beginning of the section
- Discussion: a more in-depth discussion about the novelty and comparison between the present work and other literature articles should be included
- Conclusions: future development and practical applications should be clearly stated
- Overall, several English language and grammar mistakes are present, as well as typos, and should be corrected.
Reviewer 3 Report
Having read with great interest the earlier work in this area by Ewa Gieysztor, I was hoping that this paper would be an obvious extension. I believe this is an important message you are attempting to convey - but this particular paper has a number of problems which must be addressed:
Line 1: This should just read Abstract
Lines 13-14: I think this should read: ……..are an expression of the possible malfunction …….. this is far from being a proven fact!
Lines 15-17: Please rewrite – not clear regarding the meaning
Lines 27-29: Please rewrite – not clear
Lines 54-73: Although I understand what you are trying to say here the writing style requires some work. I believe that this work is something of an extension of earlier work published by some of the authors and this should be stated clearly as it offers a good base from which to work.
Line 78: You have used the term “quenched” on numerous occasions – I feel that this is an inappropriate term – disappeared would be a better word to use.
Line 82: Strauss (startle) reflex
Lines 84-102: Not clear – perhaps a line drawing of photo here would male these descriptions more clear.
Lines 124-155: You do not discuss here when these primitive reflexes should normally disappear – I believe this to be an important point since you will need to refer to it in the results section.
Lines 156: More information required as to how this was documented.
Results section:
This needs to be completely rewritten. The way your results are presented is unclear.
Line 206: Where is Table 1?
Line 213: what is an elderly child?????
In addition I have made comments on the actual paper.
Much of the lack of clarity in this paper could be alleviated by either line-drawings/pictures or photographs.
The biggest problem is that the results of the sensory profile were missing and the fact that I was unable top actually see this linkage in your results section was a huge gap.
Your discussion and conclusion and were interesting but I feel that the level of academic English is not good enough for publication and requires much work.
